# ST(OR)$^2$: Spatio-Temporal Object Level Reasoning for Activity Recognition in the Operating Room

**Idris Hamoud**[1]                                                          IHAMOUD@UNISTRA.FR
**Muhammad Abdullah Jamal**[2]                                  ABDULLAH.JAMAL@INTUSURG.COM
**Vinkle Srivastav**[1,3]                                              SRIVASTAV@UNISTRA.FR
**Didier Mutter**[3]                                      DIDIER.MUTTER@CHRU-STRASBOURG.FR
**Nicolas Padoy**[1,3]                                                   NPADOY@UNISTRA.FR
**Omid Mohareri**[2]                                          OMID.MOHARERI@INTUSURG.COM

[1] *ICube, Universite de Strasbourg, CNRS, Strasbourg, France*

[2] *Intuitive Surgical Inc., Sunnyvale, USA*

[3] *IHU Strasbourg, Strasbourg, France*

**Editors:** Accepted for publication at MIDL 2023

## Abstract

Surgical robotics holds much promise for improving patient safety and clinician experience in the Operating Room (OR). However, it also comes with new challenges, requiring strong team coordination and effective OR management. Automatic detection of surgical activities is a key requirement for developing AI-based intelligent tools to tackle these challenges. The current state-of-the-art surgical activity recognition methods however operate on image-based representations and depend on large-scale labeled datasets whose collection is time-consuming and resource-expensive. This work proposes a new sample-efficient and object-based approach for surgical activity recognition in the OR. Our method focuses on the geometric arrangements between clinicians and surgical devices, thus utilizing the significant object interaction dynamics in the OR. We conduct experiments in a low-data regime study for long video activity recognition. We also benchmark our method against other object-centric approaches on clip-level action classification and show superior performance.

**Keywords:** OR Workflow Analysis, OR Surgical Activity Recognition, Object Level Reasoning, Human-object interaction

## 1. Introduction

Modern-day surgeries are continually evolving from an artisanal craft to a high-tech discipline. Minimally invasive surgery, particularly robotic-assisted surgery (RAS), has revamped the traditional surgical approaches, reducing postoperative recovery and hospitalization time (Sheetz KH, 2020). Still, some barriers are limiting its overall adoption. As these procedures require the clinicians to operate in a spatially constrained and technically challenging environment, it can result in poor intraoperative team communication (Schiff et al., 2016) and hindrance in the OR workflow (Catchpole et al., 2015; Kanji et al., 2021).

The recent advent of surgical data science (SDS) (Maier-Hein et al., 2022) aims to build AI-based OR assistance systems to tackle these challenges. The emerging new approaches,

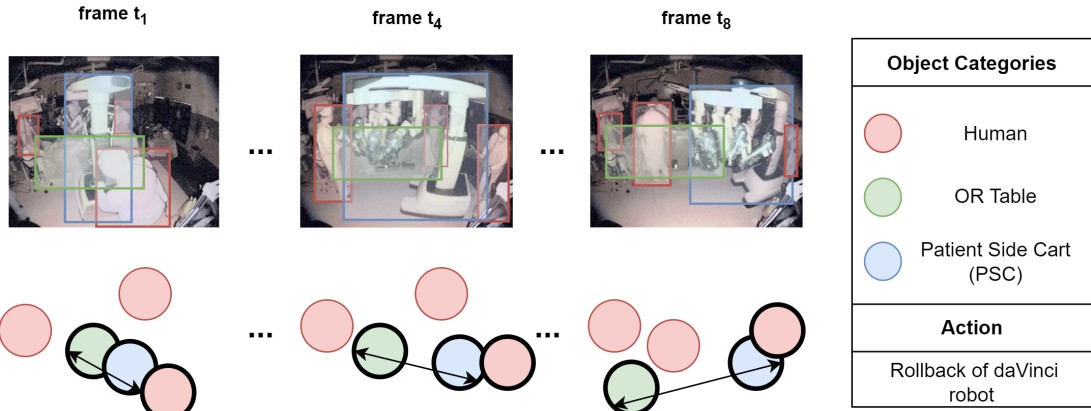

Figure 1: Example video of an action class, in this case "daVinci Rollback". Our method focuses on geometric interactions of semantically identified objects. Here the proximity of one of the clinicians with the PSC and the fact they are both getting away from the OR table is a strong clue to predict the correct action.

such as semantic scene understanding of OR (Li et al., 2020), activity analysis in robot-assisted surgery (Sharghi et al., 2020; Schmidt et al., 2021), surgical workflow recognition (Zhang et al., 2021; Kadkhodamohammadi et al., 2021; Twinanda et al., 2017b), and radiation risk monitoring during interventional surgical procedures (Rodas et al., 2016), show the potential of such systems in streamlining clinical workflow processes and supporting real-time decision-making (Vercauteren et al., 2019; Padoy, 2019; Mascagni and Padoy, 2021).

One of the core technologies developed in this direction is surgical activity recognition (SAR), which aims to segment surgical videos into phases temporally. This coarse temporal segmentation is a first step towards measuring OR efficiency for robot and resource usage, designing better workflows, and improving human-machine collaboration. Inspired by recent advances in action recognition in the general computer vision community (Carreira and Zisserman, 2017; Feichtenhofer et al., 2018; Bertasius et al., 2021; Patrick et al., 2021), SAR uses clip-based models to segment the temporal phases in surgical videos (Sharghi et al., 2020; Schmidt et al., 2021; He et al., 2022; Hajj et al., 2018; Czempiel et al., 2020; Twinanda et al., 2017a). Although powerful, these methods work on either clip-level representations or global image-based feature representations and do not explicitly embed local object-based feature representations. This lack of structured representation burdens data efficiency, requiring a model to use large-scale annotated datasets for generalization. This is a crucial concern, especially in surgical data analysis, due to the high cost and effort necessary to gather a sufficient number of well-annotated videos.

This work proposes a novel OR surgical activity recognition approach that exploits object-level dynamics. The idea of our approach bears its root in developmental psychology (Tenenbaum et al., 2011) and its adoption in computer vision (Materzynska et al., 2019). The development psychology theories agree on object-centric reasoning being a core constituent of human-level common sense. It has been shown that decomposing a scene into

meaningful units, including foreground and background objects, can benefit from the numerous advantages of abstract symbolic representation. These include data efficiency, the ability to reason over semantically defined objects, and compositionality for better generalization with less annotated data. Objects being the atomic units of physical interactions, it seems reasonable to discretize surgical scene representations at the object level.

Inspired by these, our approach employs geometrically grounded surgical scene constituents, i.e., bounding boxes of clinicians and OR objects, to build a Spatio-Temporal geometric interaction graph. As shown in Figure 1, the object-centric modeling of the surgical scene and its temporal evolution provide strong cues to predict correct surgical action. We use simple multi-layer perceptron networks to reason over the object-centric interaction graph for action recognition. This turns out to be the key to showing the generalizability of our method with less labeled data. Our experiment shows that by using as few as 5% of action labels, our approach reaches 54.2 mAP, surpassing the global feature-based baseline by 15%. The paper's contributions are two-fold:

1. We propose a new geometrically grounded object-centric approach for SAR.
2. We achieve significantly better results against the baseline methods across different percentages of labeled supervision.

## 2. Related Work

### 2.1. Surgical Video Understanding

Surgical activity recognition has been widely studied in laparoscopic videos and ophthalmological videos, mainly due to the availability of public datasets such as Cholec80 (Twinanda et al., 2017a) or Cataracts100 (Hajj et al., 2018). State-of-the-art approaches to videos capturing complete procedures all adopt two-stage training. The first stage serves as a feature extractor on individual frames, whereas the second stage encapsulates long-range temporal patterns. In TECNO (Czempiel et al., 2020) proposed to use a TCN (Lea et al., 2016) on top of ResNet18 (He et al., 2015) features for each frame to temporally segment videos of cholecystectomy. (Murali et al., 2022) recently proposed a graph-based approach for critical view of safety assessment using the geometric disposition of anatomical structures.

Our work falls into the scope of holistic operating room video understanding (Srivastav et al., 2019, 2018; Luo et al., 2018; Sharghi et al., 2020; Li et al., 2020; Chakraborty et al., 2013; Issenhuth et al., 2019). The pioneering works on this type of data have been proposed for hand hygiene compliance and human pose estimation for clinicians in the operating room. An important aspect of these datasets is privacy preservation through the use of low-resolution color (Srivastav et al., 2020), or depth (Srivastav et al., 2019) images.

In 4D-OR, (Özsoy et al., 2022) introduced a new dataset with densely annotated 3D scene graphs in an orthopedic OR. They exploit geometric interactions and annotated interactions to determine clinicians' roles.

In the work proposed by (Sharghi et al., 2020), authors introduced an OR Activity Recognition dataset for workflow monitoring, they proposed a two-stage approach with a 3D CNN coupled with an LSTM (Hochreiter and Schmidhuber, 1997). Their work was later on extended by (Schmidt et al., 2021), who proposed to use an attention layer to merge information from multiple views of the OR. Recently, (He et al., 2022) proposed an extensive

benchmark of temporal models on this same dataset. (Jamal and Mohareri, 2022) proposes an unsupervised pre-training approach using multi-modal data for OR understanding under low-data regimes. In this work, we want to address the data efficiency issues by introducing a new object-centric model that utilizes the spatial configuration of objects in the OR.

## 2.2. Object-centric Video Models

Video understanding benchmarks such as action recognition on large publicly available datasets have recently seen a significant improvement with the development of approaches based on 3D CNN (Carreira and Zisserman, 2017; Feichtenhofer, 2020; Feichtenhofer et al., 2018) and Video Transformer (Bertasius et al., 2021; Patrick et al., 2021). Those methods albeit powerful do not explicitly exploit spatio-temporal human-object interaction cues.

Recently, many approaches to make use of relational reasoning have been proposed in the video-understanding community. To spatially ground features from a 3D CNN, STRG (Wang and Gupta, 2018) proposes a class-agnostic approach using a temporal graph over densely sampled regions of interest in video frames. In STIN (Materzynska et al., 2019), the authors build an object graph using both position and category information of tracked objects to recognize actions in a compositional setting. In ORViT (Herzig et al., 2022), authors proposed a fully transformer-based approach to reason over object information in two ways: by attending to pooled object features similarly to STRG (Wang and Gupta, 2018) and by encoding object trajectories at different intermediate layers using tracked bounding boxes of objects. A detailed study of the transferability of object-centric action recognition to other downstream tasks was lately proposed by (Zhang et al., 2022).

## 3. Methods

We propose Spatio-Temporal Object-level Reasoning in the Operating Room $(\text{ST(OR)}^2)$ for OR surgical activity recognition. $\text{ST(OR)}^2$ takes as input the 2-d bounding boxes extracted from a $T$ frame-long clip subsampled by two. We use the bounding box geometric information and the semantic information about each object class to build our graph. We use our spatio-temporal object graph to reason over the objects' relative locations to recognize clip actions based on interaction dynamics. This backbone serves as the first stage of our long video segmentation, as it allows us to extract reliable features.

In the second stage, we use the features extracted from our clip-based model to train a temporal sequence model to capture long-range dependencies in the videos.

### 3.1. Object-level Representation

To train our backbone, we sample a short $T$-frame long clips from each phase of the longer OR surgical videos. Each object appearing in those frames will serve as a node of our graph. Inspired by STIN (Materzynska et al., 2019), each node will be associated with specific features grounded on the position and category of the object (cf Figure2).

**Person/Object Detection** We infer bounding boxes for all videos using two Cascade Mask RCNN (Cai and Vasconcelos, 2018) with a ConvNext backbone (Liu et al., 2022) pretrained on OR images for human and RAS-specific objects. We will use $N$ bounding

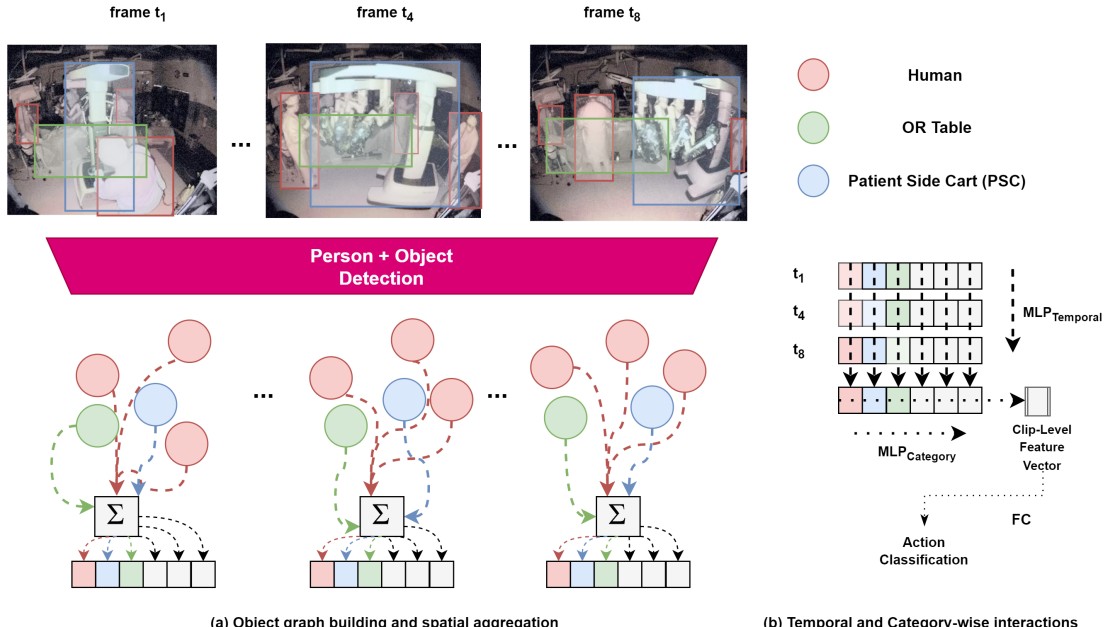

Figure 2: Architecture of our method, we first build our object graph and aggregate features category-wise. In the second step, we reason over time for each category and then we reason over categories to obtain a clip-level feature vector for action classification.

boxes for each frame of the videos. If a frame has fewer than N boxes, the remaining boxes are pad with zeros.

**Spatial Position Embedding**  We represent object position using their bounding box coordinates as a 4-d vector containing center coordinates and the width and height of the box. This 4-d vector is then forwarded to a multi-layer perceptron (MLP), to obtain a d-dimension embedding.

$$\sigma_{i,t} = MLP_{SPE}(box_{i,t}) \quad i \in \{1, \ldots, N\}, \ t \in \{1, \ldots, T\} \tag{1}$$

**Category Embedding**  We use the knowledge about the classes of objects to enhance each node's representation. Each of those $C$ classes will be associated with a d-dimension learnable embedding, that is randomly initialized from an independent multivariate normal distribution.

$$class_{i,t} = c \in \{1, \ldots, C\} \quad \kappa_{i,t} = Embed_c \quad i \in \{1, \ldots, N\}, \ t \in \{1, \ldots, T\} \tag{2}$$

Both spatial position embedding and category embedding are concatenated and passed through an MLP to obtain a fused representation for each node of the graph.

$$x_{i,t} = MLP_{Fusion}(\sigma_{i,t} || \kappa_{i,t}) \quad i \in \{1, \ldots, N\}, \ t \in \{1, \ldots, T\} \tag{3}$$

### 3.2. Spatio-Temporal Reasoning

**Category-wise Aggregation** We aggregate features of objects belonging to the same category by summing them together. This alleviates any need for tracking different instances of the same class across frames, but also discards instance-wise specific information for humans.

$$\varphi_{c,t} = \sum_{class_{i,t}=c} x_{i,t} \quad c \in \{1, \ldots, C\}, \ t \in \{1, \ldots, T\} \tag{4}$$

**Temporal-Category Interaction Module** Using the aggregated features for each object category, we first carry out temporal reasoning across categories after concatenating the $\varphi_{c,t}$ features for each frame $t$ of the clip.

$$\varphi_c = MLP_{Temp}(\varphi_{c,1}||...||\varphi_{c,T}) \quad c \in \{1, \ldots, C\} \tag{5}$$

Once we obtain a feature vector representing the temporal evolution of each object category, we perform category-wise reasoning over the concatenated features of each category to obtain a clip-level representation. We use a cross-entropy loss on the output probabilities to train our clip classification backbone.

$$\phi_{clip} = MLP_{Category}(\varphi_{c=1}||...||\varphi_{c=C}) \tag{6}$$

Our object-level representation can also easily be combined with video appearance features extracted from a 3D CNN, in our experiments we will be using I3D (Carreira and Zisserman, 2017).

### 3.3. Temporal Sequence Modeling

Following our clip-based feature extraction, each video is then represented as $v_i = \{\phi_1, ..., \phi_T\}$ with $\phi_t$ is the feature extracted from the $t^{th}$ clip. Those features can then be concatenated with the I3D features and be fed to Uni-GRU (Cho et al., 2014). This allows us to deal with long-range temporal dependencies and obtain a more robust temporal segmentation for long videos.

## 4. Experiments and Results

### 4.1. OR SAR Dataset

We demonstrate the performance of our method on the OR Surgical Action Recognition (SAR) dataset that was first introduced by (Sharghi et al., 2020). This dataset contains 400 full-length videos extracted from 103 surgical procedures. Acquisition of the videos is performed by 4 ToF cameras positioned strategically on two different carts to properly capture the full OR.

Nine activities relative to robotic system utilization are annotated on the entire dataset, and class-specific statistics are provided in (Sharghi et al., 2020). This dataset was later extended with bounding box annotations for persons and objects. Five OR-specific objects are annotated on around 19K frames across 20 full-length videos. Those activities and objects are illustrated in Figure 3.

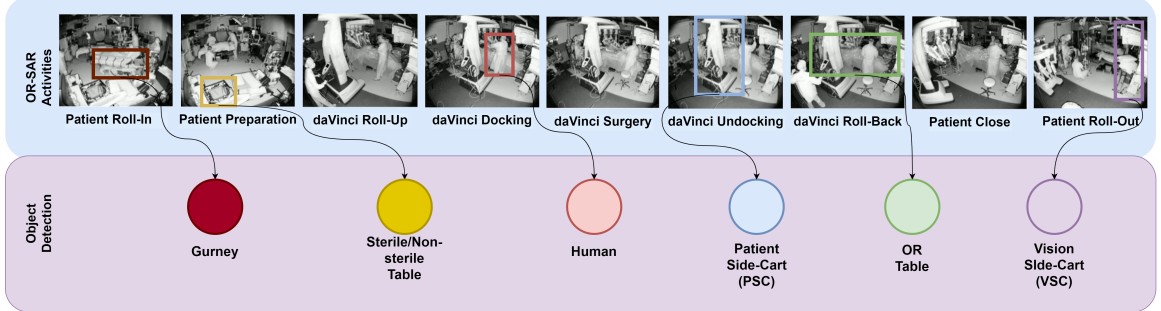

Figure 3: Overview of the OR SAR dataset, with nine clinically relevant activities and six OR-specific objects annotated.

Table 1: Results and comparison against baselines for OR surgical activity recognition on complete procedures. mAP (%) is reported across three different splits for all data fractions. We provide the mean average precision averaged across splits.

| Temporal Model | Backbone | Visual Features | 2 % | 5 % | 10 % | 20 % | 100 % |
|---|---|---|---|---|---|---|---|
| | | | **Surgical Activity Recognition** | | | | |
| Uni-GRU | I3D | ✓ | 19.7±2.8 | 39.2±1.9 | 53.5±1.5 | 79.5±0.9 | 90.7±0.6 |
| | ST(OR)$^2$ | × | 27.3±2.1 | 48.8±1.7 | 58.2±1.9 | 68.3±1.6 | 73.6±1.3 |
| | I3D + ST(OR)$^2$ | ✓ | **29.5±2.3** | **54.2±1.7** | **60.1±1.6** | **82.3±1.4** | **91.8±1.0** |

## 4.2. Data Efficiency Experiments

For the first experiment, we evaluate our method using an increasing amount of annotated data to assess if our model can learn with fewer activity labels. As shown in Table 1, ST(OR)$^2$ performs much better than the global feature-based I3D baseline (Carreira and Zisserman, 2017). Up to 10% of labeled data, ST(OR)$^2$ is better than the I3D without using any visual features. This shows the potential of encoding surgical scenes using our proposed Spatio-Temporal geometric interaction graph for OR surgical activity recognition. We further improve the performance across all percentages of labeled data when we integrate global image-based I3D features with local object-based features. We report the average mean-average precision (mAP) across three different splits, that we sampled randomly, for all data fractions.

## 4.3. Clip Classification Experiments

We evaluate our method against both global image-centric and local object-centric approaches. We choose I3D (Carreira and Zisserman, 2017), TimesFormer (Bertasius et al., 2021), and MotionFormer (Patrick et al., 2021) as the global image-centric baselines. We choose STRG (Wang and Gupta, 2018) and ORViT (Herzig et al., 2022) as the local object-centric baselines. The STRG uses global features from I3D, ORViT uses global features from MotionFormer, and our proposed ST(OR)$^2$ uses global features from I3D. We evalu-

Table 2: Results and comparison against baselines for clip-based surgical action classification. top-1 accuracy (%) is reported.

| Surgical Action Classification | | |
| --- | --- | --- |
| **Backbone** | **Object-Centric Model** | **Top-1 Accuracy** |
| I3D | $\times$ | 87.9±0.9 |
| TimesFormer | $\times$ | 85.7±0.4 |
| MotionFormer | $\times$ | 84.5±1.2 |
| I3D | STRG | 88.4±1.2 |
| MotionFormer | ORViT | 84.1±1.4 |
| $\times$ | $ST(OR)^2$ | 47.3±2.1 |
| I3D | $ST(OR)^2$ | **89.4±0.8** |

Table 3: Results of the object feature ablation study. top-1 accuracy(%) is reported with standard deviation. We cut out separately the Spatial Position Embedding (SPE) and the Category Embedding (CE).

| Feature Ablation | |
| --- | --- |
| **Node Features** | **Top-1 Accuracy** |
| No SPE | 33.1 ± 3.1 |
| No CE | 44.1 ± 2.2 |
| SPE + CE | **47.3 ± 2.1** |

ate the clip-based action classification using Top-1 accuracy. Table 2 shows that baseline approaches without using local object features perform inferior to those using local object features. The proposed $ST(OR)^2$ overperforms all other baselines.

### 4.4. Ablation Experiments

We investigate the feature representation of the nodes of our object graph, to show the importance of both spatial position and category embeddings. We run an ablation study by removing spatial position features and category embedding features, respectively. As shown in Table 3, we notice a significant 14.2% and 3.2% drop in performance when not using the spatial position features and category embedding features, respectively. This shows the strong dependence of our approach on the geometric grounding of the scene elements. The category information of the geometrically grounded objects further improves the results.

### 5. Conclusion

This paper proposes a novel geometrically grounded object-centric approach for OR surgical activity recognition. Our approach exploits the geometric layout of the clinicians and the surgical devices and builds a Spatio-Temporal graph to reason about the underlying surgical activity. We show that our object-centric approach provides superior results against the baseline methods and works particularly well with less labeled supervision. Furthermore, this work only uses a rough geometric representation of objects, i.e., bounding boxes, to

induce object awareness in visual representations. In the future, more spatially rich features like human pose estimation of clinicians could be employed to further improve OR surgical activity recognition performance.

**Informed consent** Data has been collected within an Institutional Review Board (IRB) approved the study and all participants informed consent has been obtained.

## Acknowledgments

The majority of this work was carried out during an internship by Idris Hamoud at Intuitive Surgical Inc. This work is supported by a Ph.D. fellowship from Intuitive Surgical and by French state funds managed within the "Plan Investissements d'Avenir" by the ANR (reference ANR-10-IAHU-02).

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

Table 4: Class-specific performance for object detection class-specific. mAP at IoU:0.5:0.95 (%) is reported on the testing set for our detector.

| Object Detection Performance | | | | | | |
|---|---|---|---|---|---|---|
| Method | Human | Table | Gurney | PSC | OR Table | VSC |
| Cascade MaskRCNN | 79.3 | 65.4 | 57.4 | 70.2 | 46.4 | 69.7 |

## Appendix A. Implementation Details

In our experiments, we set the number of bounding boxes used in each frame to 15. This number was chosen by taking the max number of objects per video across all videos in the training set. The dimension of the spatial position embedding and the category embedding is chosen as 1024, We train all our backbones for 50 epochs using stochastic gradient descent as our optimizer. We use a cosine policy learning rate with a warmup start learning rate of 0.01, a momentum equal to 0.9, and a weight decay of $1e-6$. Our models are trained on 4 Nvidia V-100 GPUs, using a batch size of 32 clips.

## Appendix B. Object Detection Specifics

The main limitation of our method is the weak object detector as we only rely on bounding box information to recognize clip actions. We provide per-class performance for our object detector in Table 4. We also report in the same table the performance of the human detector which is significantly superior.

## Appendix C. Class-specific performance for Action Recognition

We compare the performance of our approach without any visual features for the nine different OR surgical activities. As is shown in the confusion matrix provided in Figure 4, our approach tends to underperform in recognizing the Docking, Undocking, and Surgery phases. Those three phases are consecutive and only involve fine-grained movement of the clinician's hands to install or uninstall the instruments on the PSC. The lack of overall movement from the surgical equipment does not provide enough information to our model to correctly classify those actions. On the other hand, actions that are closely associated with the movement of surgical devices like the Roll-Up and Roll-Back of the daVinci robot give the best performance with a compelling 69% Top-1 accuracy.

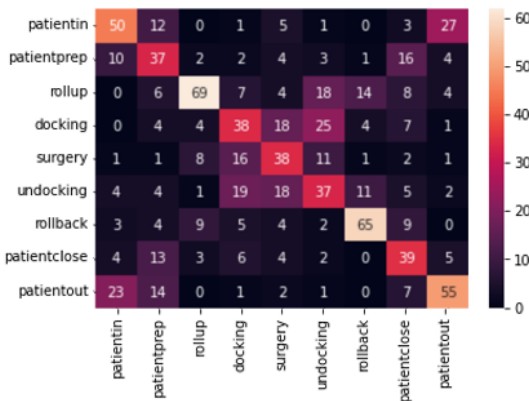

Figure 4: Confusion Matrix for OR Surgical Action Recognition.

