# OpenReview forum: "ST(OR)$^2$: Spatio-Temporal Object Level Reasoning for Activity Recognition in the Operating Room"
_MIDL.io/2023/Conference — MIDL 2023 Poster_

### Official Review · Reviewer_Lh5B · 2023-02-01

**Confidence:** 4
**Preliminary Rating:** 4
**Recommendation:** Oral, Poster

**Summary:**

Automatic detection of surgical activities is important for developing AI-based tools for tackling specific challenges. The existing method requires a large amount of labeled data, which is often not available for the Operating Room (OR) setting. To solve this problem, the author proposes to focus on the geometric arrangement between clinicians and surgical devices via a simple yet effective spatial and category embedding. To this end, the method not only outperforms the baseline by a large margin under the data scares setting, but also achieves better performance when the full dataset is used.

**Strengths:**

Training an action recognition model under a sample-efficient setting is important for some medical applications such as OR. This could be especially challenging when the data is video data, which is known for being data-hungry. To tackle this problem, the author leverages the geometric arrangement between the objects that are important in OR. To this end, a sample-efficient and object-centric embedding is proposed. The efficiency has been demonstrated via sufficient experiments for both the low-data regime and the full set, for the long video activity recognition and clip-level action classification.

**Weaknesses:**

This paper is generally well-written with convincing results. I am just curious how the following setting may be and would like to discuss with the author:

I3D is pretrained on the Kinetic dataset, which is not object-centric, sometimes it puts more emphasis on the static background than the motion and interaction between objects. I am wondering how the model will perform if 1) we finetune the I3D model on our dataset in a lightweight way and 2) we pre-train the visual model or use the already pretrained model on a more motion-centric dataset, e.g. Something-Something dataset. Would that improve the visual performance under the low data regime?

**Deanonymize Review:**

no

**Detailed Comments:**

See above

**Paper Type:**

both

**Questions To Address In The Rebuttal:**

Some intuitive discussions are welcomed in terms of activity recognition for OR domain concerning the question that I raised in the weakness section. Otherwise, I think the paper is currently in a mature status.

---

### Official Review · Reviewer_MRNG · 2023-02-02

**Confidence:** 4
**Preliminary Rating:** 3

**Summary:**


This manuscript proposes ST(OR)2, a learning-based framework for operating room activity recognition. ST(OR)2 uses existing image-based detection network (i.e., Mask RCNN) and video-activity network (i.e., I3D) to process the video data captured from ToF cameras. The geometric interaction module is based on STIN. Experiments are conducted on the in-house dataset with additional 2D annotations. The baseline is primarily I3D. Ablations on data-scarce settings are presented, showing better results for ST(OR)2 than I3D.

**Strengths:**

The paper studies an interesting problem – video activity recognition in OR. The paper further annotates a significant amount of data in 2D (19K frames, 20 videos), which can be used for video activity recognition.

**Weaknesses:**

- Lack of motivation for technical approach – While it is interesting to adapt STIN to OR activity recognition, the assumption of STIN is violated and thus raising the quality of the results.

- A mismatch between the claimed contribution and experiments – Some experiments weaken the claim of the proposed network.

- Lack of description of the experiment setup.


**Deanonymize Review:**

no

**Detailed Comments:**

Lack of motivation of technical approach:
- In STIN, the spatial interaction has been assumed to be between instances, not categories. And their results have been shown on a single hand manipulating objects, without any category-level ambiguities (such as two hands).
- However, in the submitted paper, activities are inferred based on categories but not instances Sect.3.2 (Category-wise Aggregation). The instance features are summed, and thus losing identity information. Based on the figures, there can be multiple humans in the scene, which in theory introduces ambiguities. Is there proof that the networks can handle multiple instances? How does the network reason about the “geometric” relationship between objects if the instance information is lost?

A mismatch between the claimed contribution and experiments:
- The proposed method is claimed to reason “geometrically” and perform “significantly better across different levels of supervision”, particularly in data-scarce settings. Supposedly, geometric reasoning boosts performance and builds towards a solution to activity recognition.
- However, in 100% supervision setting, ST(OR)2 underperforms I3D by a large margin: $73.6 \pm 1.3$ vs $90.7 \pm 0.6$. I3D without “geometric” reasoning performs already well and the “geometric” reasoning of ST(OR)2 itself is actually quite insufficient. Furthermore, ST(OR)2 + I3D only improves marginally than I3D: $91.8 \pm 1.0$ vs $90.7 \pm 0.6$. Is the improvement significant? It appears to me that all the effort of extracting bounding boxes does not provide many benefits.
- It has been shown that in the data-scare setting (2% - 10%), ST(OR)2 performs better than I3D. However, I am not sure what readers can gain from this improvement. With 10% data, the best network is only at around 50-60 top-1. With 100% data, the best network is at around 90 top-1. It looks like a large volume of data is needed regardless to solve the activity recognition problem. How is such data-scarce result motivating? Unless ST(OR)2 works in an unsupervised manner, thus scalable with the dataset collection, which is not demonstrated in the manuscript.

Lack of description of experiment setup:
- What is the train/val/test split of the video? Are data split across video or all frames?
- What is the distribution of the activities (how many frames per activity)? What are the respective top-1 accuracies for each activity?
- How are top-1 acc calculated? Are they average across all frames? This ties back to the previous question. If there were a single activity that dominates all video sequences, networks could just learn that single output.

Misc
- “Sample efficiency” can be ambiguous on its own. I would suggest something like “data efficiency”.
- I would suggest clarifying the input to the network from the beginning. For a long time, I assumed the bounding boxes are 3D because of the ToF cameras used. However, I think what is actually used is the intensity images of the ToF cameras.
- I would suggest replacing “Surgery video” with “operating room surgery video” or “operating room video”. “Surgery video” may refer to videos observed by endoscopes.
- Why not use depth information from ToF? This in theory gives more meaning to "geometric" reasoning.
- Fig 1, the dashed circle is not defined anywhere. I think this is OR table, but using the wrong notation from the legends and the rest of the figures.
- Eqn 3/5/6, what is “||”? Concatenation?
- Eqn 6, when is $\phi_{clip}$ used? When there is no I3D?
- Sect 3.3, what are $\phi_1$, $\phi_2$, $\phi_T$? Are these from Eqn 4? If so, the $\phi_{c,t}$ notation is not correct. It should have been $\phi_t$ without $c$.


**Paper Type:**

methodological development

**Questions To Address In The Rebuttal:**

- Theoretical reasoning and empirical evidence show that instance information is not lost, and thus does not suffer from category-level ambiguities
- Explanations on the results, casting doubts on if "geometric reasoning" is helpful and if "data-scarce" settings are meaningful
- Additional descriptions of the experiment setup
- Fixes some of the typos and flow of the paper for better readability

---

### Official Review · Reviewer_LNXb · 2023-02-06

**Confidence:** 4
**Preliminary Rating:** 4
**Recommendation:** Oral, Poster

**Summary:**

1. The authors suggest a new approach to surgical activity recognition that is based on geometry and objects. Since objects are the basic units of physical interactions, it makes sense to represent surgical scenes as collections of objects.
2. The proposed approach was tested against baseline methods and was found to produce significantly better results across a range of labeled supervision percentages.




**Strengths:**

1. The central concept of using geometric information about the components of a surgical scene is innovative and practical. Potentially, the approach leverages the compositionality of surgical scenes, allowing for better generalization to new and unseen activities with less annotated data. These benefits make the proposed approach a novel and valuable solution for recognizing surgical activities in the operating room.
2. The figures presented in the paper are visually appealing and effectively convey the information. They are clear and concise, making it easy for readers to understand the results and conclusions of the research. Additionally, the paper is well-written and well-structured, making it an enjoyable and accessible read. The language used is clear and concise, making it easy to follow the logic and arguments presented in the paper.
3. The experiments described in the paper are well-designed and executed. They provide a comprehensive evaluation of the proposed approach for surgical activity recognition and its performance compared to other methods. The results of these experiments appear to be promising, indicating that the proposed approach is an effective solution for recognizing surgical activities in the operating room

**Weaknesses:**

1. The authors acknowledge that the main limitation of the proposed method is the weak object detector, as it relies solely on bounding box information to recognize surgical activities. In some cases, this may result in inaccurate recognition, particularly when the bounding boxes are not correctly aligned with the objects they represent. To further improve the proposed method, it would be beneficial for the authors to provide more detail on the failure cases and perform a deeper analysis.
2. This analysis could include identifying common scenarios where the method struggles and examining the reasons for these failures. The authors could also consider ways to improve the object detector, for example, by incorporating additional information, such as object masks or object features, to make the recognition more robust and accurate. An in-depth analysis of the failure cases and potential solutions would significantly improve the robustness and effectiveness of the proposed method.





**Deanonymize Review:**

no

**Paper Type:**

methodological development

**Questions To Address In The Rebuttal:**

1. At this pre-rebuttal stage, I would rate this paper as '4: weak accept'. Although the central concept of using geometric information in surgical activity recognition appears to be novel, and the experimental results are promising, my main concerns are related to the lack of sufficient detail and further analysis of failure cases.
2. I recognize that the paper presents an innovative approach and demonstrates promising results, but I believe that further analysis of the limitations and failure cases would significantly strengthen the argument and impact of the research. I am eager to review the authors' responses to my concerns and to see how they address these limitations in their revision. I believe that with a more comprehensive analysis and additional details of failure cases, this paper has the potential to make a significant contribution to the field of surgical activity recognition.

---

### Meta-Review · Area_Chair_dytb · 2023-02-24

**Recommendation:** Accept (Poster)
**Confidence:** 5

**Metareview:**

The manuscript enjoyed active discussion during the rebuttal phase, with both reviewers and authors contributing constructively to improve the paper.

There is reviewer consensus that the method is novel and the idea of leverging geometry is relevant. Sample efficiency is also mentioned as a strength.

The major weaknesses largely pertain to the performance of the method in larger data regimes, where the baselines seem to perform quite well while the proposed method underperforms. Additional experiments have been added to investigate related issues, and while the object detector required for this work seems to continue to pose challenges, the corresponding reviewer was - to some degree - satisfied with the additions. I am, however, in agreement that these additional experiments and considerations need to be added to the manuscript.